# New 5-Substituted SN38 Derivatives: A Stability Study and Interaction with Model Nicked DNA by NMR and Molecular Modeling Methods

**DOI:** 10.3390/ijms242417445

**Published:** 2023-12-13

**Authors:** Elżbieta Bednarek, Wojciech Bocian, Jerzy Sitkowski, Magdalena Urbanowicz, Lech Kozerski

**Affiliations:** National Medicines Institute, Chełmska 30/34, 00-725 Warsaw, Poland; wo.bocian@gmail.com (W.B.); j.sitkowski@nil.gov.pl (J.S.); m.urbanowicz@nil.gov.pl (M.U.); l.kozerski@nil.gov.pl (L.K.)

**Keywords:** camptothecin derivatives, SN38, DNA model, NMR, molecular modeling, docking

## Abstract

The new 5-substituted SN-38 derivatives, 5(*R*)-(N-pyrrolidinyl)methyl-7-ethyl-10-hydroxycamptothecin (**1**) and its diastereomer 5(*S*) (**2**), were investigated using a combination of nuclear magnetic resonance (NMR) spectroscopy and molecular modeling methods. The chemical stability, configuration stability, and propensity to aggregate as a function of concentration were determined using ^1^H NMR. The calculated self-association constants (*K_a_*) were found to be 6.4 mM^−1^ and 2.9 mM^−1^ for **1** and **2**, respectively. The NMR experiments were performed to elucidate the interaction of each diastereomer with a nicked decamer duplex, referred to as **3**. The calculated binding constants were determined to be 76 mM^−1^ and 150 mM^−1^ for the **1**–**3** and **2**–**3** complexes, respectively. NMR studies revealed that the interaction between **1** or **2** and the nicked decamer duplex occurred at the site of the DNA strand break. To complement these findings, molecular modeling methods and calculation protocols were employed to establish the interaction mode and binding constants and to generate molecular models of the DNA/ligand complexes.

## 1. Introduction

In recent years, numerous new derivatives of camptothecin (CPT) have been patented and described in the literature [1,2,3,4,5], reflecting the continued significance of chemotherapy as a standard medical approach against cancer. Topoisomerase I (Topo I) inhibitors play a crucial role in this therapeutic strategy [4,6,7]. Specifically, Topo I inhibitors from the camptothecin family (as shown in Figure 1), such as Topotecan (Hycamtin™) and Irinotecan (Camptosar™, known as CPT-11), are employed in clinical cancer treatments, particularly against various solid tumors, including breast (BRCs) and colorectal cancers (CRCs). It is worth noting that these camptothecin derivatives, known for their effectiveness in Topo I inhibition, primarily form ternary complexes involving nicked DNA (DNA/Topo I/Inhibitor) [6]. As the drug is bound within the cleavage site, it prevents DNA relegation by misaligning the DNA end, which is normally required to attack the phosphotyrosyl bond. That prevents the rotation of the cleaved strand around the helical axis and, in result, cannot relegate the cleaved strand to reestablish the duplex DNA [8]. According to this mechanism, the effectiveness of anticancer drugs should be related to the strength of the drug’s binding to the Topo I/DNA complex. However, when it comes to binding isolated oligomer DNA, these compounds demonstrate relatively poor affinity [9].

In the literature [3,10,11], various derivatives of the parent camptothecin have been discussed. These derivatives substituted at position C5 with heteroatoms, bulky amines, esters, and amides have demonstrated moderate to good biological activity, while substituents at positions C7, C9, C10, and C11 do not significantly influence the geometry of a ternary complex, and substitution in rings D and E have the potential to adversely affect biological activity [3].

One of the key novel aspects investigated in this research is the role of individual enantiomers substituted at position C5, their contribution to stabilizing the molecular complex DNA/inhibitor, and the resultant impact on the biological activity of these compounds. We proposed a hypothesis that this substituent could potentially act as an anchor, situated within the minor groove and interacting with the active surface of an enzyme [4]. Our preliminary results regarding mono C5-substituted compounds [12] support this hypothesis, as their cytotoxicity is even higher than that of compounds expected to covalently bind with DNA [13]. Notably, this particular issue has not been extensively addressed in recent literature due to the scarcity of data on the biological activity of pure enantiomeric 5-substituted camptothecin derivatives with known stereochemistry at C5 [14]. Therefore, this study represents an effort to determine whether the C5 substituent, containing the nucleophilic nitrogen, has the potential to influence the binding strength within the molecular complex DNA/inhibitor.

Medication safety is a crucial aspect, and it is closely linked to specificity and the strength of binding to its target. In this context, Irinotecan cannot be viewed as the first-choice medication but, rather, as a last-resort option due to its metabolite SN38’s potential to cause severe and often fatal diarrhea, ultimately leading to death. This is one of the reasons why we are investigating the role of substituents at C5 as an additional factor in enhancing the affinity of the entire molecule to the DNA/Top I complex. The kinetic nature of this process requires a strong interaction to influence the on/off timing of a complex. It is widely acknowledged that the camptothecin core intercalates with DNA bases in a nicked DNA, a phenomenon supported by our previous works [9,13,15]. However, less is known about the impact of substituents at C5 [16] or the interactions between inhibitor’s substituents and Topo I [4]. Introducing a substituent at the prochiral center C5 creates a steric factor that can influence the strength of binding to nicked DNA, which serves as the target for Topo I inhibitors.

The potential to alkylate nitrogen bases within the DNA nick by a prospective inhibitor presents a path toward developing safety for oncological patient pharmaceutics. Aiming towards advancing new camptothecins into the early phases of clinical studies, we embarked on an endeavor to elucidate the pharmacological role of the newly synthesized diastereomeric derivatives in the process of Topo I inhibition. This led us to investigate the compounds shown in Figure 2, with the hope of gaining deeper insights into the mechanisms underlying their interaction with a model of nicked DNA, as depicted in Figure 3. The absolute configuration at the C5 carbon atom of compounds **1** and **2** was determined through electronic circular dichroism (ECD), and in vitro biological assays were conducted on various cancer cells (colon HT29, breast MCF7, blood HL-60, and lung A549) and normal cells CRL1790 (see Table 1) [12]. As these compounds exhibited favorable half-maximal inhibitory concentrations (IC_50_), and of equal importance, both diastereomers demonstrated significantly lower toxicity towards non-neoplastic cells when compared to SN38, we decided to study them.

In this paper, NMR studies were conducted on novel derivatives of SN38 that had been substituted at the C5 position with a (N-pyrrolidinyl)methyl group—specifically, **1** (5*R*,20*S*) and **2** (5*S*,20*S*). The chemical and stereochemical stability over time, as well as the aggregation process, were investigated. Furthermore, NMR-derived insights were provided on the interactions between compounds **1** (5*R*,20*S*) and **2** (5*S*,20*S*) with a model nicked DNA molecule, the nicked decamer duplex **3**. Notably, this model included a G–C base pair within the nick, mimicking one of the aspects of a wild-type nicked DNA [9]. It has been previously demonstrated that the camptothecin core exhibits a preference for binding to the G–C base pair in a natural octamer [15]. Additionally, molecular modeling techniques were employed to predict the theoretical molecular geometry of both complexes involving the studied diastereomers and the nicked decamer duplex **3.**

This study represents an effort to determine whether the C5 substituent, containing nucleophilic nitrogen, has the potential to influence the binding strength within the molecular complex DNA/inhibitor.

## 2. Results and Discussion 

The experimental ^1^H and ^13^C NMR chemical shifts for compounds **1** and **2** in D_2_O/DMSO-*d*_6_ solution*,* (90%/10%, pH 3) are presented in Appendix A. For further clarity, the ^1^H NMR spectra of **1** and **2** are depicted in Figure 4. Notably, the chemical shifts of proton signals for compounds **1** and **2** exhibited distinct differences, making it feasible to distinguish the configuration at C5 based on the ^1^H NMR spectra. Another distinguishing factor between diastereoisomers is the retention time, R_f_ (see the HPLC chromatograms of compounds **1** (R_f_: 16.3 min) and **2** (R_f_: 17.6 min) in Appendix A).

### 2.1. Chemical and Stereochemical Stability of ***1*** and ***2*** in Water

The property of chemical and stereochemical stability in aqueous environments is crucial for any potential medication to reach its biological target without significant structural changes. NMR methodology was employed to assess the decomposition process of compounds **1** and **2**. Changes in the signal intensity observed in the ^1^H NMR spectra of the tested compounds over time, along with the appearance of additional signals, typically serve as indicators of an ongoing decomposition process. To evaluate stability, compounds **1** and **2** were dissolved in a D_2_O/DMSO-*d_6_*, 90%/10% mixture with a pH of 3. Subsequently, the ^1^H NMR spectra were measured in a quantitative regime at 25 °C over a 45-day period. These stability studies were conducted in a solvent composition often employed in studies involving cells or animals.

The experimental results revealed no significant changes in signal intensity above the level of experimental error. Moreover, no additional signals appeared in the spectra (see Appendix A). The results of those experiments confirmed the stability of both diastereomers under experimental conditions. Furthermore, compounds **1** and **2** retained their stable absolute configuration at the C5 and C20 carbon atoms throughout the incubation period in the solution.

### 2.2. Aggregation Studies of ***1*** and ***2*** in Buffered Water Solution

To assess the impact of the geometry of both diastereomers on their propensity to aggregate in an aqueous solution, which can be associated with their bioavailability, we performed dilution experiments while monitoring them using ^1^H NMR and diffusion ordered spectroscopy (DOSY).

The ^1^H NMR spectra were acquired over a relatively limited range of concentrations (ranging from 10 μM to 1.0 mM) due to the restricted solubility of the tested compounds. The ^1^H NMR data obtained from the dilution experiments are presented in Table 2 and are shown in Figure 5. It is noteworthy that all proton signals, with the exception of H23 and H26/H27 for compound **1** and H25/H28 and H19 for compound **2**, were shifted to lower frequencies. For compound **1**, the most significant shielding changes were observed for the aromatic protons H12, H11, and H14, as well as for the aliphatic protons H5 and H24 (ranging from 120 Hz to 185 Hz). In contrast, the chemical shift changes for the aliphatic protons H17 in ring E and the protons of the groups attached to ring B (H23), ring E (H18 and H19), and ring C (H26/H28) were much smaller, ranging from 20 Hz to 57 Hz (see Table 2). Similar effects were observed for compound **2** (see Table 2), with the exception that the change in H24 proton shielding was notably reduced (approximately 140 Hz for compound **1** compared to around 40 Hz for compound **2**). Large changes in the chemical shifts of protons H12 and H14 during self-association suggest strong stacking interactions in a uniform mode, specifically face-to-face interactions with rings A.

The average values of the self-association constants *K_a_*, calculated based on dilution experimental data, were 6.4 ± 2.2 mM^−1^ for **1** and 2.9 ± 1.6 mM^−1^ for **2**, as shown in Table 2. It is worth noting that the error of the self-association constants for both compounds was relatively significant. Furthermore, an unexplained substantial deviation of the self-association constants calculated using specific protons, such as H12 in compound **1** and H23 and H19 in compound **2** (marked in red in Table 2), was observed.

Nonetheless, it can be inferred from the observed chemical shift changes and the calculated association constants that **1** exhibits a stronger self-association compared to **2**. This discrepancy can be elucidated by examining the structural features of the compounds under study. In the case of **1**, both bulky substituents, (*R*)C5-CH_2_-PYR and (*S*)C20-C_2_H_5_, are located on the same side of the camptothecin core, facilitating close stacking of the molecules. Conversely, in the case of **2**, the two large groups are positioned on opposite sides of the camptothecin core, making monomer stacking less favorable.

The self-association of both diastereomers was confirmed by diffusion experiments performed on samples with selected concentrations, as depicted in Table 3. These experiments revealed that, in both cases, the value of the diffusion coefficient, Di, decreased with higher concentrations, indicating the occurrence of a self-aggregation process.

### 2.3. Binding of ***1*** or ***2*** with Nicked Decamer Duplex ***3*** Characterized by DOSY, ^1^H NMR, and NOESY Spectra

The NMR method has been used for decades to study the phenomenon of interactions between molecules. It provides insights into the structure of formed complexes and offers estimates of the binding affinity between molecules. Typically, on the NMR timescale, the exchange between uncomplexed and complexed species occurs so rapidly that a single set of signals is observed in the NMR spectra for each molecule involved in the interaction. The positions of these signals are the weighted average of the signal positions of the uncomplexed and complexed forms, with the weighting factors being the molar fractions of the respective forms.

#### 2.3.1. DOSY Experiment of Derivatives **1** and **2** with Nicked Decamer Duplex **3**

DOSY is a valuable NMR experiment for the quantitative analysis of interactions between molecules in solution [17,18]. It is particularly useful for studying the affinity of smaller molecules, such as compounds **1** or **2** (with a molecular weight, *M_W_*, of approximately 475 Da), to much larger molecules, like the nicked decamer duplex **3** (with a molecular weight of around 6887 Da). The diffusion coefficient of a small compound bounded to a larger host molecule is significantly smaller than that of its uncomplexed form in solution.

Given the different geometries of both diastereomers, one might anticipate different affinities for the model nicked DNA. DOSY experiments were performed for compounds **1** or **2**, both in the absence and presence of **3**. The binding affinity of **1** and **2** with nicked decamer duplex **3** was calculated based on diffusion coefficients measured in equimolar solutions. Only the signal of the CH_3_ group at C19 was used to determine the diffusion coefficients of **1** or **2** in the presence of **3** (these signals do not overlap with the signals of **3**) to ensure the accurate calculation of the binding constant in this experiment. The DOSY spectra are depicted in Figure 6, and the data are given in Table 4.

The calculated binding constants for complexes **1**–**3** and **2**–**3** were determined to be 76 and 150 mM^−1^, respectively (see Table 4). These calculations were performed based on the assumption of a 1:1 interaction model not accounting of self-association, which seems to be appropriate, particularly given that the calculated self-association constants for both compounds are much smaller. It is worth noting that the calculated binding constants are at least one order of magnitude higher than the binding constants previously determined for other camptothecin derivatives substituted at the C9 [9,15,19] or at the C9 and C5 positions [16]. This somewhat unexpected result highlights a notably strong interaction between the nicked decamer duplex **3** and compounds **1** and **2**. 

##### ^1^H NMR and NOESY Experiments of **1** and **2** with Nicked Decamer Duplex **3**

The nuclear Overhauser spectroscopy (NOESY) experiments were recorded for solutions of **1** and **3** or **2** and **3** to investigate the interactions between compounds. Unfortunately, no intermolecular cross-peaks between **3** and compounds **1** or **2** were observed in the NOESY spectra, making it impossible to determine intermolecular distances between protons in the formed complexes. Therefore, NOESY spectra were solely used as a tool for the assignment of proton signals in the process of determining chemical shift changes induced by the interactions of the molecules under investigation. On the other hand, a significant broadening of proton signals in the ^1^H NMR spectra for the solutions of **1** and **3** or **2** and **3** compared to the spectra of the free compounds **1** or **2** was observed, commonly attributed to dynamic phenomena. This effect was clearly observed for several nonoverlapping signals. For instance, in the case of compound **1,** proton signals from methyl groups 19-CH_3_ and 23-CH_3_, originally located at 0.98 ppm and 1.58 ppm, respectively, appeared as triplets. Under the interaction of compound **1** with **3,** they shifted to 0.95 and 1.29 ppm, respectively, and became very broad singlets. Similarly, the signal for group 18-CH_2_ at 2.00 ppm, originally a quartet, shifted to 1.91 ppm and became a broad singlet. The H12 proton signal was broadened and shifted from 7.21 ppm to 6.94 ppm. The low-frequency shifts observed in these groups suggest that the camptothecin core is stacking inside the nick, surrounded by a shielding cone of aromatic rings. 

The analysis of the NOESY spectra for solutions **1** and **3** or **2** and **3** showed significant changes in the proton shielding of both compounds **1** and **2,** as well as **3**. The chemical shift changes (Δδ) of the proton signals of **1** and **2** under interactions with **3** are presented in Table 5. For both compounds, pyrrolidine protons H25/28 and H26/27 exhibited no changes in shielding, while all other proton signals experienced a shift towards a lower frequency. The values of the chemical shift changes for the protons of both compounds **1** and **2** were similar, albeit slightly higher for **2**. Significant chemical shift changes were particularly observed for the proton signals of groups 22-CH_2_, 5-CH, 11-CH, 9-CH, and 12-CH for both diastereomers. We also observed changes in shielding for the DNA unit protons. Table 6 presents data regarding the changes in the H6/H8 and H1’ signal proton shielding of the nicked decamer duplex **3** upon interaction with compounds **1** or **2**. Most of the observed chemical shift changes fell within the experimental error. Notably, significant changes in shielding (exceeding 0.02 ppm and highlighted in red) were detected for the proton signals of DNA units located within the gap of **3**, specifically the H1′ protons of T5, G6, C15, and A16, as well as the H6/H8 protons of the G6, T7, and A16 units. The induced chemical shift changes from both compounds were generally of the same order but, in most cases, slightly greater for compound **2**. Figure 7 shows fragments of the three overlapped NOESY spectra, reflecting changes in the equilibrium state between the interacting species under investigation. The spectrum of **3** is shown in green, while the spectra of the equimolar **1** and **3** or **2** and **3** solutions are depicted in red. The blue spectra represent solutions **1** and **3** or **2** and **3** with an excess of ligands.

It is clearly seen that changes in DNA signals upon the ligand complexation predominantly occur in the G6–C15 and T5–A16 pairs, flanking the two faces of a nick. This strongly supports the conclusion that the ligand specifically binds inside the nick. Furthermore, most changes tend to have lower frequencies, suggesting stacking interactions between the ligand and DNA aromatic bases, aligning with the modeled structures (as discussed in Section 2.4).

### 2.4. Molecular Modeling of Complexes of Both Diastereomers

#### 2.4.1. Molecular Docking Analysis

Computational docking studies were conducted to predict the binding modes of our camptothecin derivatives (compounds **1** and **2**) in the model nicked DNA/Topo I complex. The open-source AutoDock software package was employed for this study, utilizing AutoDock-GPU, the CUDA-accelerated version of AutoDock 4.2.6 [20]. The two X-ray structures available in RCSB (PDB IDs: 1K4T [21] and 1T8I [22]) representing the ternary TPT/DNA/Topo I and CPT/DNA/Topo I complexes were utilized as receptors after removing the ligand components (TPT or CPT). In the initial attempt, the standard docking procedure was employed with a rigid receptor and flexible ligands using the Lamarckian genetic algorithm with an empirical free energy scoring function. The results of this docking procedure are detailed in Table 7 under the “X-ray single structure” column. Additionally, the results of docking calculations were also presented for several other reference compounds [23], such as camptothecin (CPT), topotecan (TPT), and SN38. The best docking poses for all the ligands closely resembled the conformations of the ligands (TPT and CPT) in the X-ray structures. However, the calculated binding energies for compounds **1** and **2** using this method were generally unimpressive.

To enhance the precision of the docking calculations in the next calculation attempt as the receptor model, the structures acquired during 500 ns explicit solvent molecular dynamic calculations (MDs) were used (the same as in our previous publication [23]). The application of molecular dynamics simulations enables an evaluation of receptor sidechain and backbone movements within the complex during docking, allowing, in principle, the generation of novel conformations. Two molecular dynamics trajectories were executed for the 1K4T and 1T8I X-ray starting structures, with 1000 structures uniformly recorded during each MD run. Subsequent to the removal of the TPT or CPT ligands, the recorded structures were integrated into the docking calculations. The results are presented in Table 7 under the “1000 MD-Derived Structures” column. Notably, the calculated binding energies in this instance appeared to be more accurate, better aligning with the experimental IC_50_ toxicity data to cancer cells (see Table 1).

The previous docking calculations clearly exhibited an artificial bias towards conformations in line with the X-ray structures. To address this concern, new molecular dynamics calculations were performed for ternary complexes compounds **1** and **2**/DNA/Topo I. Accordingly, starting from the X-ray structure of the TPT/DNA/Topo I complex (PDB ID: 1K4T), the TPT ligand was replaced with compound **1** or **2**. However, the new ligands were placed in all four possible distinct poses relative to the receptor. For each of the eight resulting complexes, two separate molecular dynamics simulation trajectories were conducted, yielding a total of sixteen 500 ns trajectories. As before, 1000 structures were uniformly recorded during each MD run for every trajectory. For the 8000 receptor structures thus obtained (after ligand removal), docking calculations were then performed with the use of compounds **1** and **2**. The best docked structures were subsequently classified into four distinct structural families via cluster analysis, analogous to those in the MD calculations. 

Family structure 1 aligned with the X-ray structures, family structure 2 encompassed structures where the ligand was rotated 180 degrees around an axis perpendicular to the artificial plane of the molecule, family structure 3 comprised structures with a 180-degree ligand rotation about a long axis parallel to the artificial plane of the molecule, and family structure 4 represented structures with a 180-degree ligand rotation about both axes described above (see Figure 8).

For each family of structures, 50% of the structures with the highest energies were excluded. For the remaining structures, the mean binding energies and average inhibition constant values were calculated. The results for the reference compounds come from publication [23], and they were subjected to a similar docking procedure as for compounds **1** and **2**. However, due to the high labor intensity of the calculations, the molecular dynamics calculations for these compounds were abandoned and replaced by docking to the MD structures obtained for the compound investigated in publication [23]. It should be noted that the docking calculations for these reference compounds were inherently somewhat less robust. The results are presented in Table 8.

The calculations indicate that compounds **1** and **2** exhibit significantly stronger binding to the receptor in comparison to the reference compounds, suggesting a promising potential for the future application of these compounds. In this scenario, the computational analysis favors the formation of the complex involving compound **1** with family structure 4 and compound **2** with family structure 3. In contrast, the reference compounds consistently align with the preferred family structure 1, aligning well with the existing knowledge, including X-ray structures [7,24].

However, it is important to note that these calculated docking binding energies do not encompass the effect of entropy. Following the approach presented by Ruvinsky, one might suppose that the entropy could be proportional to the size of the structure cluster (CS) [25]. Yet, even when incorporating this consideration, our conclusions remain largely unchanged. Notably, the entropy contribution for TPT, CPT, and SN38 consistently exhibits the highest magnitude within family structure 1. For compounds **1** and **2**, comparable sizes of clusters for some families of structures are observed.

Consequently, we can deduce that, for the ternary complex compound **1**/DNA/Topo I, family structure 4 emerges as the preferred conformation while acknowledging the potential existence of a dynamic equilibrium with family structure 1. Similarly, in the case of the ternary complex compound **2**/DNA/Topo I, family structure 3 appears as the favored arrangement, yet the possibility of a dynamic equilibrium with family structure 2 cannot be discounted.

#### 2.4.2. MM-PBSA and MM-GBSA Calculations of the Full Ligand/DNA/Topo I Ternary Complex

With the available molecular dynamics simulations for the ternary complexes involving compounds **1** and **2**, we proceeded to calculate the molecular mechanics ligand binding energies using the Poisson–Boltzmann or generalized Born and surface area continuum solvation methods (MM-PBSA and MM-GBSA). These methods have proven successful in replicating and rationalizing experimental observations, as well as enhancing the outcomes of virtual screening and docking studies [16,19,23]. Nevertheless, it is important to acknowledge that these methods entail several simplifications and approximations, including the omission of conformational entropy and information regarding the quantity and free energy of water molecules within the binding site. One of the prominent challenges is the requirement for accurate sampling of the conformational space, which often necessitates lengthy molecular dynamics simulations. In our specific case, we conducted simulations spanning 500 ns, a duration within our computational capacity (comprising 25,000 structural samples for each trajectory); however, there is a possibility that this duration may not be sufficient [23]. The outcomes of these calculations are tabulated in Table 9, with data for the reference compounds TPT and CPT sourced from our previous work [23]. Despite the inherent limitations, we can confidently reaffirm that compounds **1** and **2** exhibit considerably stronger binding affinity to the receptor in comparison to the reference compounds. Notably, the preferred conformations for the compounds **1** and **2**/DNA/Topo I complex are those corresponding to family structures 4 and/or 1.

#### 2.4.3. Hydrogen Bonds Analysis in the Full Compounds **1** and **2**/DNA/Topo I Ternary Complex

Intermolecular hydrogen bonds (HBs) assume a critical role in ligand–receptor interactions, often constituting a substantial energetic factor contributing to binding energy [26]. The statistics of these HB bonds are presented in Appendix A and in Figure 8. The table contains statistics separately for each family of structures and separately for structures obtained from docking calculations and molecular dynamics simulations. The provided statistics encompass the average length of specific hydrogen bonds, along with their occurrence frequency, indicating the percentage of structures wherein a particular bond that is observed.

By analyzing the hydrogen bond data, it can once again be inferred that, in the ternary complex compound **1**/DNA/Topo I, family structure 4 emerges as the favored conformation. Similarly, in the case of the ternary complex compound **2**/DNA/Topo I, family structure 3 is observed as the preferred arrangement.

## 3. Materials and Methods

### 3.1. Chemical Substrates

The nicked DNA decamer duplex **3** was purchased from International DNA Technologies (IDT, 1710 Comercial Park, Coralville, IA 52241, USA) and purified by the filtering of water solution on a membrane of 3 kDa. The compound 5(*R*)-(N-pyrrolidinyl)methyl-7-ethyl-10-hydroxycamptothecin (**1**) and its diastereomer 5(*S*) (**2**) were obtained as formate salts using one-pot synthesis. The compounds were separated and purified by HPLC, as described earlier [12].

### 3.2. HPLC 

HPLC was performed using a HPLC system from Shimadzu USA Manufacturing, Inc. (Canby, OR, USA), consisting of a low-pressure gradient flow LC-20AT pump, a DGU-20A online solvent degasser, a SPD-M20A photodiode array detector, a SIL-10AF sample injector, and a FRC-10A fraction collector. Data were monitored using a Shimadzu LabSolution system. HPLC was performed on a Phenomenex Gemini 5 µm NX-C18 110 Å 250 × 4.6 mm column with a mobile phase system of CH_3_CN/aqueous 0.1% HCOOH at a flow rate of 1 mL min^−1^ using the following gradient: 5% CH_3_CN from 0 min to 30% CH_3_CN at 15 min to 50% at 20 min. Chromatography was monitored using UV detection at a wavelength of 260 nm giving retention times of 16.3 min (**1**) and 17.6 min (**2**).

### 3.3. Sample Preparation

Samples of **1** and **2** for the stability experiments were prepared in D_2_O/DMSO-*d*_6_, 90%/10%, with the internal standard sodium 3-trimethylsilyltetradeuteriopropionate (TSPA-*d*_4_), pH 3. 

For the aggregation and binding studies, the samples were dissolved in D_2_O buffer (25 mM NaCl/25 mM K_3_PO_4_) with internal standard TSPA-*d*_4_, pH 6. Compounds of the camptothecin family exist in lactone and carboxylate forms at pH 7 [9]. Therefore, the pH was adjusted to 6 to avoid formation of the carboxylate form of compounds **1** and **2** in solution, which have a very weak binding affinity to DNA. 

### 3.4. NMR Experiments

The NMR spectra were recorded at 10 °C or 25 °C using a Varian VNMRS-500 spectrometer (Varian, Inc., NMR Systems, Palo Alto, CA, USA) operated at 499.8 and 125.7 MHz for the ^1^H and ^13^C measurements, respectively. All experiments were run using the standard Varian software (VnmrJ version 3.1 A software from Varian, Inc., NMR Systems, Palo Alto, CA, USA). The spectrometer was equipped with an inverse ^1^H{^31^P–^15^N} 5 mm Z-SPEC Nalorac IDG500-5HT probe (Nalorac Corp., Martinez, CA, USA) with an actively shielded *z*-gradient coil to give a maximum gradient strength of 61.1 G cm^−1^.

The NMR spectra were referenced using TSPA-*d*_4_ as the internal reference. The concentrations of **1**, **2**, and **3** in the tested solutions were determined against quantitatively added TSPA-*d*_4_. One-dimensional proton spectra were acquired in conditions that assured quantitative measurements using 16–2048 scans (depending on the concentration), with a 30° pulse width and a relaxation delay of 10 s.

Two-dimensional experiments were performed under the following conditions:

NOESY: spectral widths 6000 Hz in both dimensions, 1024 complex points in *t*_2_, 400 complex points in *t*_1_, 16–64 scans per increment, relaxation delay 1 s, and mixing time 300 ms.

^1^H-^13^C heteronuclear single-quantum correlation (HSQC): spectral widths 6000 Hz in F2 and 21,600 Hz in F1, 1024 complex points in *t*_2_, 512 complex points in *t*_1_,16 scans per increment, relaxation delay 1s, and ^1^*J*(C,H) = 146 Hz.

^1^H-^13^C heteronuclear multiple-bond correlation (HMBC): spectral widths 6000 Hz in F2 and 23,880 Hz in F1, 1024 complex points in *t*_2_, 256 complex points in *t*_1_, 64 scans per increment, relaxation delay 1s, and ^n^*J*(C,H) = 8 Hz.

One-shot [27] DOSY spectra: 16–1024 transients, 16 dummy scans, diffusion time (Δ) 120 ms for **1** or **2** and 200 ms for octane **3** and bound species, total diffusion encoding gradient duration (δ) 2.8 ms, and 16 values of the diffusion-encoding gradients incremented from 6 to 50 G/cm in such steps that the strength of the next gradient was equal to the previous gradient squared. Other parameters included the following: a sweep width of 12,000 Hz, 48 k data points, an acquisition time of 2.0 s, and a relaxation delay of 2.0 s. Processing was carried out using VARIAN VNMRJ software, with the option of correction for spatially nonuniform pulsed field gradients. 

### 3.5. Calculating the Self-Association Constants of ***1*** and ***2***

The concentration dependence of chemical shifts of the **1** or **2** protons signals was obtained from the NMR experiment. A 0.79 mM solution of **1** and a 1.06 mM solution of **2** in phosphate buffer (D_2_O, 25 mM NaCl/25 mM K_3_PO_4_) at pH 6.0 were diluted stepwise with the same buffer down to about 0.013 mM. The one-dimensional ^1^H NMR spectra were measured for these solutions in conditions assuring the quantitative measurements of a concentration. The concentration in the solutions was measured by comparing the integral of chosen proton signals of **1** or **2** with the integral of the TSPA-*d*_4_ signal. The experimental data were used for evaluation of the association constant *K_a_* of **1** or **2** (L) by the isodesmic model [28], which assumes that L associates to form stacks, with a single self-association constant, *K_a_*. The data from the dilution experiment were used to fit Equation (1) [29]:(1)$$\Delta \delta_{\mathrm{obs}}=\Delta \delta_{\max}\;\cdot \;K_{a}\cdot \;\left[L_{0}\right]\cdot \;{\left(\frac{2}{1+\sqrt{4\;\cdot \;K_{a}\cdot \;\left[L_{0}\right]+1}}\right)}^{2}$$
where Δδ_obs_ = δ_mon_ − δ_obs_ means the change in the observed average chemical shift for proton signals of L at different concentrations, δ_obs_ means the observed average chemical shifts of proton signals of L at different concentrations, δ_mon_ means the chemical shifts of protons signals of L at infinitely low concentrations (in monomer form), Δδ_max_ refers to the maximal chemical shift change between monomer and oligomer, [L_0_] means the total concentration of L, and *K_a_* means the association constant.

### 3.6. Calculating Binding Constants from the Diffusion Coefficients

The binding constants (*K_a_*) of the complex DNA·L were estimated by the analysis of the diffusion coefficients of nicked decamer duplex **3** (DNA), compounds **1** or **2** (L), and DNA·L complex as a function of the DNA and L concentrations [30]. 

Assuming that the binding equilibrium in the NMR experiment is established very quickly (dynamic equilibrium method) and the complexes exist with a stoichiometry of 1:1, the binding behavior can be described with the following mathematical model:DNA+L ↔DNA·L
(2)$$K_{a}=\frac{\left[\mathrm{DNA}\cdot L\right]}{\left[\mathrm{DNA}\right]\left[L\right]}$$
where L is compound **1** or **2**; DNA is nicked decamer duplex **3**; DNA·L is a 1:1 complex; [L], [DNA], and [DNA·L] are the equilibrium concentrations of L, DNA, and the DNA·L complex, respectively; and *K_a_* is the corresponding binding constants.

In the case in which the exchange rate between the uncomplexed and complexed species is fast on the NMR timescale, the observed diffusion coefficients (D (m^2^ s^−1^)) are a weighted average of the diffusion coefficients of the uncomplexed and complexed species, where the weighting factors are the relative population sizes of the respective species. Thus, the observed diffusion coefficients may be expressed as
(3)$$D_{\mathrm{OBS}-L}={\mathrm{MF}}_{L}D_{L}+\left(1-{\mathrm{MF}}_{L}\right)D_{\left[\mathrm{DNA}\cdot L\right]}\;$$
(4)$$D_{\mathrm{OBS}-\mathrm{DNA}}={\mathrm{MF}}_{\mathrm{DNA}}D_{\mathrm{DNA}}+\left(1-{\mathrm{MF}}_{\mathrm{DNA}}\right)D_{\left[\mathrm{DNA}\cdot L\right]}$$
where D_OBS-L_ and D_OBS-DNA_ are the observed averaged diffusion coefficients for L and DNA measured in the solution containing both L and DNA, D_L_ and D_DNA_ are the diffusion coefficients for uncomplexed L and uncomplexed DNA, MF_L_ and MF_DNA_ are the molar fractions of uncomplexed L and uncomplexed DNA in the solution containing both molecules, and D_[DNA·L]_ is the diffusion coefficient for the DNA·L complex.

The *K_a_* can be also expressed as
(5)$$K_{a}=\frac{\left[\mathrm{DNA}\cdot L\right]}{\left(C_{\mathrm{DNA}}-\left[\mathrm{DNA}\cdot L\right]\right)\left(C_{L}-\left[\mathrm{DNA}\cdot L\right]\right)}\;$$
where C_DNA_ and C_L_ are the initial concentrations of DNA and L.

The unknown complex concentration can be calculated from equations
(6)$$\left[\mathrm{DNA}\cdot L\right]=\left(1-{\mathrm{MF}}_{\mathrm{DNA}}\right)C_{\mathrm{DNA}}$$
(7)$$\left[\mathrm{DNA}\cdot L\right]=\left(1-{\mathrm{MF}}_{L}\right)C_{L}$$

In the case where the DNA molecule is much larger than the L, it can be assumed that the diffusion coefficient of the DNA·L complex is the same as that of the DNA molecule:(8)$$\;D_{\left[\mathrm{DNA}\cdot L\right]}≅D_{\mathrm{OBS}-\mathrm{DNA}}\;$$

By combining Equations (3), (5), (7), and (8), the *K_a_* can be determined.

This formal treatment of the data includes a simplification that may cause the results to be affected by an error.

### 3.7. Molecular Dynamics Calculations 

The human DNA topoisomerase I complex with camptothecin and the 22 base pair DNA duplex (PDB ID: 1T8I and 1K4T) were used as the starting model structure for the calculations. The PDB X-Ray structures were carefully inspected and cleaned up, e.g., a nonstandard TGP nucleotide was converted to the standard nucleotide with a guanine G base, and all boundary nucleotides were capped. The compound **1** and **2** structures were manually docked to the nick in the receptor as needed in all four possible stacking orientations. This resulted in eight structures, which were then subjected to molecular dynamics calculations (MDs). All MD calculations were carried out using the AMBER 14 suite of the programs [31]. The electrostatic potential (ESP) charges were obtained for the ligand compounds, the nonstandard phosphate-containing amino acid PTR linker, and the DNA linkers as needed by the HF/6-31G* calculations using the Gaussian 09 program [32]. Next, the RESP charges were calculated by charge fitting with the multi-conformational procedure of the antechamber module implemented in AMBER. The missing GAFF force field parameters were obtained using the parmchk module. Each complex was neutralized by adding Na^+^ cations and then solvated by TIP3 water molecules with a spacing distance of about 15 Å around the system surface, creating a periodic box. All complexes were subjected to molecular dynamics simulations (MDs) using the pmemd.cuda AMBER 14 module with NVIDIA GPU acceleration and a mixed ff12SB-GAFF force field. The particle mesh Ewald (PME) [33] method was used to treat long-range electrostatic interactions, and a 10 Å cutoff was applied to the nonbonded Lennard–Jones interactions. The SHAKE algorithm was applied to constrain all the bonds involving hydrogen atoms, and a 2 fs time step was used in the dynamics simulation. First, the systems were minimized in two stages: the first stage restrained the atomic positions of the solute and only relaxed the water, and the second stage released the restraint and allowed all atoms to relax (both with 10,000 minimization steps). Next, the systems were slowly heated to 300 K using an NVT ensemble and 1,000,000 steps with the Langevin dynamics temperature control (gamma_ln = 1.0). Then, the systems were carefully equilibrated at NPT ensemble simulations at 1 bar pressure with gamma_ln = 5.0. The equilibrations lasted until the system reached a converged density value, usually for 10–20 ns. Finally, the NPT (constant pressure/temperature) production molecular dynamics were run for 500 or 1000 ns of simulations.

#### 3.7.1. Calculating the Binding Free Energies (Enthalpies) Using the MM-PBSA and MM-GBSA Methods [34]

The MD trajectories were uniformly sampled, yielding 25,000 samples for each trajectory. The water and Na^+^ cations were stripped, and the binding free energies (enthalpies) were calculated according to the following equations:
ΔG°_Bind,Solv_ = ΔG°_Bind,Vacuum_ + ΔG°_Solv,Complex_ − (ΔG°_Solv,Ligand_ + ΔG°_Solv,Receptor_)

The solvation free energies were calculated by solving either the linearized Poisson–Boltzmann or generalized Born equation for each of the three states (this provides the electrostatic contribution to the solvation free energy) and adding an empirical term for the hydrophobic contributions:ΔG°_Solv_ = ΔG°_electrostatic,ϵ_ = 80 − ΔG°_electrostatic,ϵ_ = 1 + ΔG°_hydrophobic_

ΔG°_Vacuum_ was obtained by calculating the average interaction energy between receptor and ligand and taking the entropy change upon binding into account.
ΔG°_Vacuum_ = ΔE°_MM_ − TΔS°
where G°_Bind,Solv_ is the free energy of the binding of the solvated molecules; G°_Bind,Vacuum_ is the binding free energy in vacuum; G°_Solv,Complex_, G°_Solv,Ligand_, and G°_Solv,Receptor_ are the solvation free energy for the complex, ligand, and receptor molecules; G°_Electrostatic_ is the electrostatic solvation free energy; G°_Hydrophobic_ is the hydrophobic (nonpolar) solvation free energy; E°_MM_ is the molecular mechanic energy; T is the temperature; and S° is the entropy.

The entropy contribution in our calculations was neglected because of a comparison to states of similar entropy. All free energy calculations were carried out using the mm_pbsa.pl script from AmberTools.

#### 3.7.2. Cluster Analysis

The CPPTRAJ module implemented in the AMBER package was used for the cluster analysis. During cluster analysis, similar conformations were identified and grouped together. During clustering analysis, the kmeans clustering algorithm was used. The RMSD of heavy atoms was used as a distance metric calculated only for the ligand derivative and the neighboring two DNA base pairs on each side for the structures from the PBSA/GBSA calculations and one DNA base pair for the structures from the PM7 calculations. The clustering procedure was repeated several times, and each time, the low-population strange structures were gradually removed. Finally, for each system, the most populated clusters were obtained. For each cluster, the average energies were calculated (PBSA/GBSA or PM7), and the most representative structures were determined.

#### 3.7.3. Docking Calculations

Docking calculations were performed using the AutoDock suite program [35]. AutoDock is a computational docking program based on the empirical free energy force field and rapid Lamarckian genetic algorithm search method [36,37]. To overcome the rigid receptor simplification in AutoDock, the docking studies were performed on ensembles of structures of complexes obtained from molecular dynamic simulations. For each complex, a set of 1000 uniformly sampled structures was used. The coordinate files were prepared using AutuDockTools, and pre-calculations of the atomic affinities were performed using AutoGrid. Both programs are parts of the AutoDock suite. The standard docking parameters were used together with the GPU accelerated version of AutoDock4.2.6. The fifty possible binding conformations were each time-generated and analyzed, including sorting in family structures, using home build scripts.

#### 3.7.4. The Hydrogen Bonds Analysis

The hydrogen bonds were identified and analyzed using proprietary scripts and the UCSF Chimera ver. 1.14 program [38]. The chimera program was also used to visualize the chemical structures. Appendix A lists hydrogen bonds only with a population greater than 1%, at least for one of the calculation methods.

## 4. Conclusions

In this contribution, we have described preliminary development studies on novel camptothecin derivatives, specifically two optically pure diastereomers of the 5-substituted SN38 core: 5(*R*)-(N-pyrrolidinyl)methyl-7-ethyl-10-hydroxycamptothecin (**1**) and its diastereomer, 5(*S*) (**2**). Our research has revealed that both compounds exhibit exceptional chemical and stereochemical stability. Their stability, combined with favorable IC_50_ values determined for various cancer cell lines and low toxicity to non-neoplastic cells when compared to SN38, positions them as highly promising compounds for further extended preclinical studies. Furthermore, self-association properties, important for drug bioavailability, were established and characterized by association constants.

Because these compounds belong to a group of Topo I inhibitors, their interactions with the model of the biological target, the nicked decamer duplex, **3**, were elucidated by NMR experiments. The DOSY experiments indicated strong interactions of both **1** and **2** with **3**. The calculated binding constants for the **1**–**3** and **2**–**3** complexes were found to be 76 mM^−1^ and 150 mM^−1^, respectively. These values were at least an order of magnitude higher than those observed for other camptothecin derivatives substituted at the C9 or C9 and C5 positions. Notably, the most significant chemical shift changes of the proton signals of **3** induced by interactions with **1** or **2** occurred in the base pairs G6–C15 and T5–A16, which flank the nick, as well as the adjacent base pairs T7–A14 and T4–A17. Finally, it is worth mentioning that compound **2** exhibited a slightly higher affinity to **3** than **1**. 

Further investigation of multiple conformations through extended molecular dynamics simulations allowed the classification of the structures into distinct families. Notably, compounds **1** and **2** exhibited stronger binding to the receptor compared to the reference compounds (camptothecin, topotecan, and SN38). Family structures 3 and 4 appeared to be the preferred conformations for compounds **1** and **2** in the ternary complexes, while the reference compounds aligned with family structure 1. Moreover, the MM-PBSA and MM-GBSA calculations supported the robust binding affinity of compounds **1** and **2** to the receptor, with preferred conformations matching those identified through docking and molecular dynamics simulations. Finally, the hydrogen bond analysis underscored the importance of family structures 3 and 4 in the compound **1** and compound **2**/DNA/Topo I complexes, emphasizing their role in the energetically favorable ligand–receptor interactions. 

These findings collectively contribute to a deeper understanding of the binding modes and potential therapeutic relevance of the investigated camptothecin derivatives in cancer treatment.

## Figures and Tables

**Figure 1 ijms-24-17445-f001:**
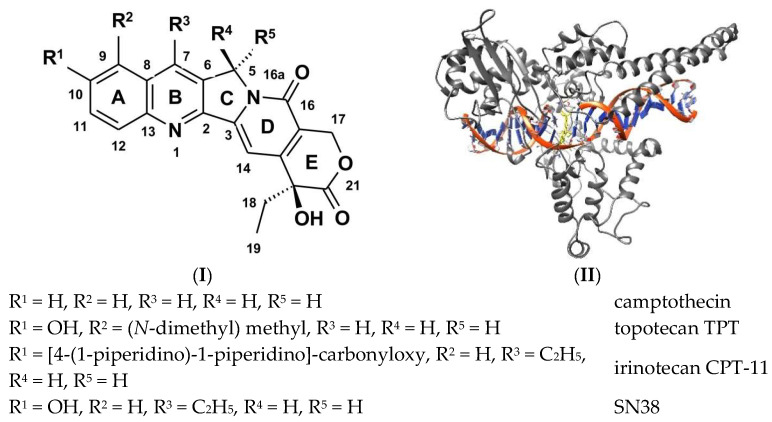
The structure of main camptothecin derivatives. Absolute configuration at C20–(*S*)–(**I**) and the structure of the ternary complex: DNA/Topo I/Inhibitor—(**II**).

**Figure 2 ijms-24-17445-f002:**
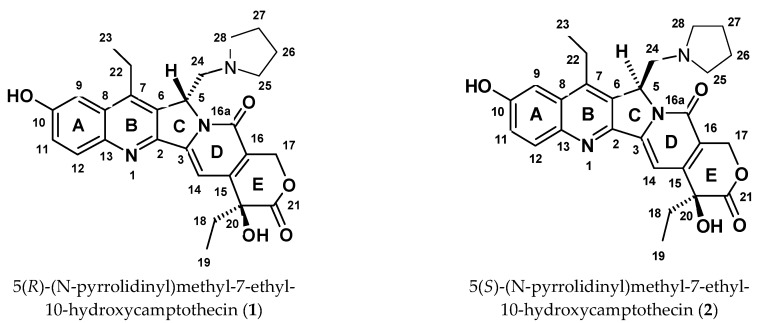
Atomic numbering and the spatial structures of compounds **1** and **2**.

**Figure 3 ijms-24-17445-f003:**
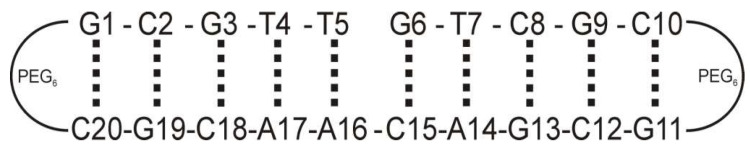
Schematic representation of a biological target of Topo I inhibitors, a nicked DNA duplex as a model decamer with a PEG(6) tether at both ends of a duplex, **3**, with a molecular weight of 6886.8 g/mol.

**Figure 4 ijms-24-17445-f004:**
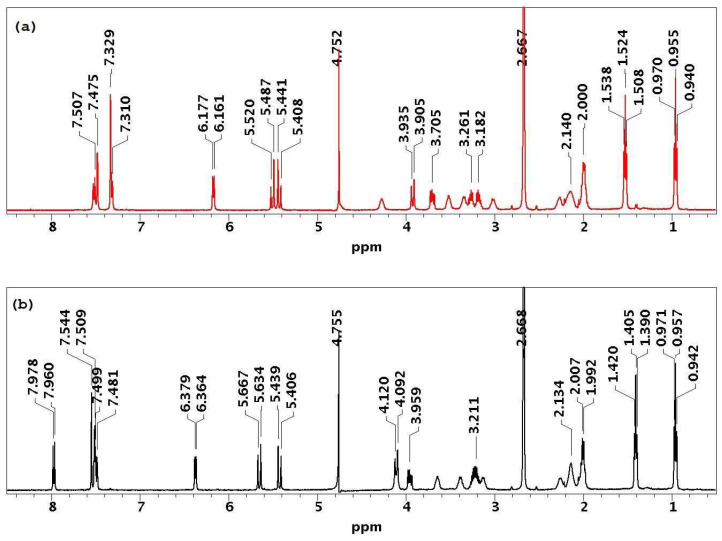
The ^1^H NMR spectra of diastereomers (**a**) **1** and (**b**) **2** in D_2_O/DMSO-*d*_6_, 90%/10% solution, pH 3, temp. 25 °C.

**Figure 5 ijms-24-17445-f005:**
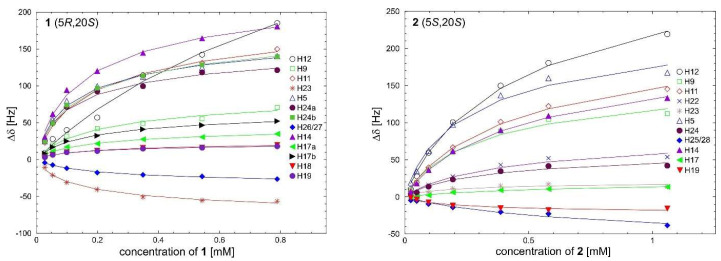
The chemical shift changes Δ*δ* (Hz) of the proton signals of **1** (left) and **2** (right) as a function of concentration (mM) (D_2_O buffer, 25 mM NaCl/25 mM K_3_PO_4_, TSPA-*d*_4_, pH 6.0, temp. 10 °C). Positive values correspond to a low-frequency shift.

**Figure 6 ijms-24-17445-f006:**
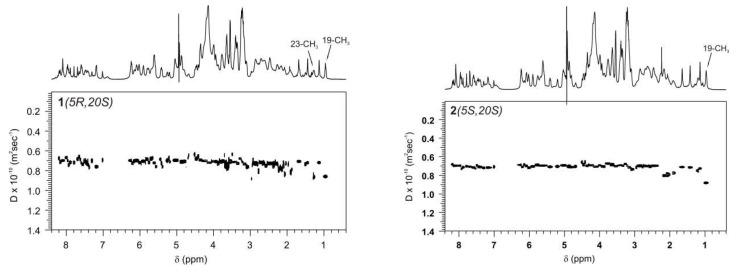
The DOSY spectra of a 1:1 molar ratio of **1** and **3** or **2** and **3** solutions in D_2_O buffer, 25 mM NaCl/25 mM K_3_PO_4_, TSPA-*d*_4_, pH 6.0, temp. 10 °C.

**Figure 7 ijms-24-17445-f007:**
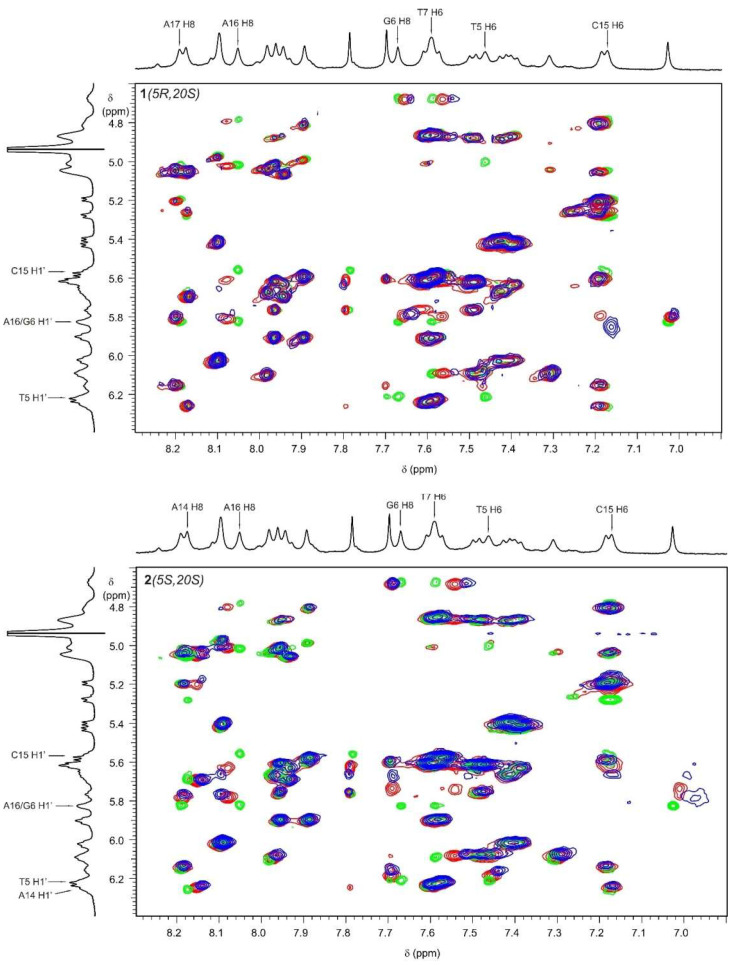
The selected region of the three overlapped NOESY spectra of the solutions in D_2_O buffer, 25 mM NaCl/25 mM K_3_PO_4_, TSPA-*d*_4_, pH 6.0, and a temperature of 10 °C. The vertical axis covers the range of H6/H8 proton signals, while the horizontal axis includes the range of H1’ protons of the nicked decamer duplex, **3**. The ^1^H NMR spectrum of **3** is presented as a projection spectrum. The units of **3** showing significant chemical shift changes resulting from the interaction with both **1** and **2** are depicted. Upper panel: Cross-peaks marked in green come from **3** at a concentration of c = 0.6 mM, marked in red from an equimolar solution (c = 0.6 mM) of **1** and **3,** marked in blue from a **1** and **3** solution at triple the excess of **1** (c_1_ = 0.6 mM and c_3_ = 0.2 mM). Bottom panel: Cross-peaks marked in green come from **3** at a c = 0.6 mM concentration, marked in red from the equimolar (c = 0.6 mM) solution of **2** and **3,** marked in blue from the **2** and **3** solution at double the excess of **2** (c_2_ = 0.6 mM and c_3_ = 0.3 mM).

**Figure 8 ijms-24-17445-f008:**
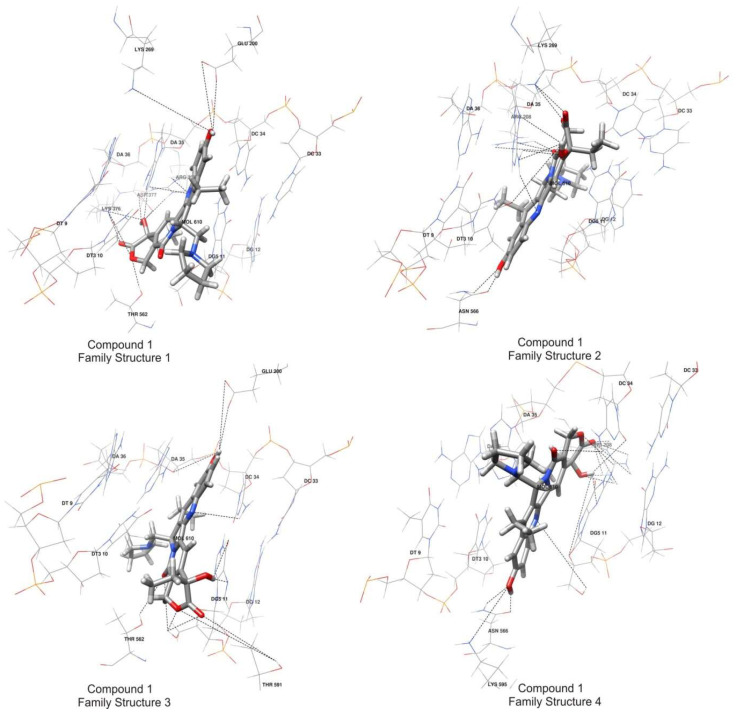
Structures representing the main clusters for each family structure obtained in the MD simulations. Only ligand-interacting fragments of the receptor are shown. Dashed black lines indicate possible hydrogen bonds for a given family of structures, the same as in Appendix A.

**Table 1 ijms-24-17445-t001:** Experimental bioassay results [12].

	IC_50_ (µM/L)				
Compound	Colon HT29	Breast MCF7	Blood HL-60	Lung A549	Normal Cells CRL1790
**1** (5*R*,20*S*)	0.598	0.810	0.011	0.204	24.440
**2** (5*S*,20*S*)	0.493	0.906	0.007	0.272	21.830
Irinotecan	6.85	14.10	4.24	26.69	90.25
SN38	0.020	0.660	0.002	0.028	4.980

**Table 2 ijms-24-17445-t002:** The chemical shift changes, Δδ ^1^ (Hz), of the proton signals in **1** (5*R*, 20*S*) and **2** (5*S*, 20*S*) in D_2_O buffer (25 mM NaCl/25 mM K_3_PO_4_, TSPA-*d*_4_) for different concentrations **c** (mM), pH 6.0, temp. 10 °C and the self-association constants *K*_a_ (mM^−1^).

Δ*δ* ^1^ for 1													
c_1_	Ring A			Ring B	Ring C				Ring D			Ring E	
	H12	H9	H11	H23	H5	H24a	H24b	H26/H27	H14	H17a	H17b	H18	H19
0.013	0.0	0.0	0.0	0.0	0.0	0.0	0.0	0.0	0.0	0.0	0.0	0.0	0.0
0.027	8.9	7.2	22.8	−10.8	25.0	25.1	24.6	−4.4	30.7	5.5	8.2	2.8	3.1
0.053	27.8	19.6	48.6	−21.1	54.2	49.2	49.2	−7.4	61.7	11.3	16.96	6.5	6.4
0.099	39.9	31.7	71.4	−31.1	79.3	71.1	73.8	−11.9	94.2	17.05	24.93	9.9	9.8
0.199	56.8	41.9	94.2	−40.7	99.2	92.4	96.8	−17.7	120.0	21.46	32.3	11.9	11.5
0.349	114.2	49.2	112.2	−50.8	112.1	99.4	113.6	−20.8	144.52	27.5	41.0	15.8	14.8
0.543	142.3	55.3	130.9	−55.5	124.6	118.1	128.5	−23.0	164.2	30.37	46.58	16.2	16.0
0.789	185.0	70.5	149.8	−56.5	140.2	121.2	139.6	−26.6	180.3	34.7	52.0	19.8	18.0
*K_a_*	** 0.554 **	4.536	5.770	8.007	9.122	10.012	7.438	5.302	6.952	5.989	5.719	6.183	7.478
Δδ_max_ ^2^	745.9	111.9	232.5	−87.7	200.0	176.3	211.2	−42.7	274.7	53.8	82.2	29.9	26.8
the average value of *K_a_*			6.4 ± 2.2 6.9 ± 1.6 (without H12)										
**Δ*δ* ^1^ for 2**													
**c_2_**	**Ring A**			**Ring B**		**Ring C**			**Ring D**			**Ring E**	
	**H12**	**H9**	**H11**	**H22**	**H23**	**H5**	**H24a**	**H25/H28**	**H14**	**H17a**		**H19**	
0.012	0.0	0.0	0.0	0.0	0.0	0.0	0.0	0.0	0.0	0.0		0.0	
0.024	10.5	8.3	7.3	0.0	2.0	19.2	3.6	−4.8	6.7	0.0		−1.3	
0.048	28.1	19.3	19.5	4.2	3.4	34.4	6.0	−5.4	18.0	0.0		−3.6	
0.096	59.5	36.8	39.5	16.2	6.5	61.8	13.8	−9.6	36.0	2.3		−7.4	
0.194	100.4	60.4	66.7	27.5	11.4	97.3	23.4	−14.0	61.5	6.2		−11.9	
0.387	149.8	88.6	101.2	43.2	14.9	136.7	34.5	−20.5	89.8	9.0		−15.2	
0.580	180.4	104.7	122.3	52.0	17.2	159.9	41.5	−22.8	109.0	10.7		−17.7	
1.060	219.2	112.1	145.3	53.7	13.7	167.2	42.1	−38.5	133.2	13.4		−15.7	
*K_a_*	1.851	2.950	1.907	2.153	** 6.322 **	3.871	2.930	1.029	1.869	1.161		** 5.573 **	
Δδ_max_ ^2^	449.4	207.6	297.1	112.5	24.8	289.4	80.1	−90.8	271.1	33.2		−27.4	
the average value of *K_a_*		2.9 ± 1.6 2.2 ± 0.9 (without H23 and H19)											

^1^ Δδ = *δ_at lowest concentration_* − *δ_at given concentration_*; positive values denote a low-frequency shift, (Hz); ^2^ Δδ_max_—calculated maximum chemical shift changes related to the dimerization/association process.

**Table 3 ijms-24-17445-t003:** The diffusion coefficients, Di × 10^−10^ (m^2^s^−1^) for different concentrations c (mM) for the studied diastereomers **1** and **2** in D_2_O buffer (25 mM NaCl/25 mM K_3_PO_4_), pH 6.0, temp. 10 °C.

1		2	
c (mM)	Di × 10^−10^ (m^2^s^−1^)	c (mM)	Di × 10^−10^ (m^2^s^−1^)
0.78 0.59 0.10 0.05	1.78 ± 0.05 1.81 ± 0.10 1.97 ± 0.10 2.12 ± 0.10	1.06 0.05	1.97 ± 0.05 2.31 ± 0.10

**Table 4 ijms-24-17445-t004:** The DOSY data ^1^ for the binding of **1** or **2** with nicked decamer duplex **3** in D_2_O buffer (25 mM NaCl/25 mM K_3_PO_4_), pH 6.0, temp. 10 °C.

	Concentration (mM)		D_OBS_	D_OBS-3_	MF_comp_	*K_a_* ± 0.3
	c	c_3_	×10^−10^ (m^2^ s^−1^)	×10^−10^ (m^2^ s^−1^)		(mM^−1^)
**1**	0.60	0.60	0.93 ± 0.05	0.74 ± 0.05	0.86 ± 0.05	76
**2**	0.60	0.60	0.87 ± 0.05	0.71 ± 0.05	0.90 ± 0.05	150

^1^ c and c_3_ are the concentrations of **1** or **2** and **3**, respectively; D_OBS_ is the measured diffusion coefficients for **1** or **2** in the presence of **3**; D_OBS-**3**_ is the measured diffusion coefficient for **3** in the presence of **1** or **2**; MF_comp_ is the calculated molar fraction of the **1**–**3** or **2**–**3** complexes; *K_a_* is a binding constant. For the measured diffusion coefficients for the uncomplexed species of **1** and **2**, see Table 3 (Di at 0.05 mM was used for the calculations); for **3** (0.60 mM), it is equal to 0.75 ± 0.05 × 10^−10^ (m^2^ s^−1^). Compounds **1** and **2** in the presence of **3** are in a fast exchange of free and bound forms, and the diffusion coefficients D_OBS_ and D_OBS-**3**_ presented in the table are a weighted average of the diffusion coefficients of the free and bound species.

**Table 5 ijms-24-17445-t005:** The experimental ^1^H NMR chemical shifts, *δ* (ppm), for **1** and **2** free and in the presence of **3** in D_2_O buffer (25 mM NaCl/25 mM K_3_PO_4_ + TSPA-*d*_4_), pH 6.0, temp = 10 °C, and chemical shift changes, Δδ ^1^ (ppm), of proton signals of **1** and **2** induced by their interaction with **3** (c_1_ = c_2_ = c_3_ = 0.6 mM).

	δ Free 1	δ 1 and 3	Δδ ^1^	δ Free 2	δ 2 and 3	Δδ ^1^
19-CH_3_	0.976	0.952	0.024	1.017	0.968	0.049
23-CH_3_	1.576	1.278	** 0.298 **	1.393	1.200	** 0.193 **
18-CH_2_	2.007	1.913	0.094	2.048	1.916	0.132
22–CH_2_	3.200	2.821	** 0.379 **	3.126	2.647	** 0.479 **
22-CH_2_	3.279	2.949	** 0.330 **	3.186	2.919	** 0.267 **
24-CH_2_	3.672	3.546	0.126	4.007	3.759	** 0.248 **
24-CH_2_	3.926	3.671	** 0.255 **	4.103	3.759	** 0.344 **
17-CH_2_	5.437	5.325	0.112	5.454	5.306	0.148
17-CH_2_	5.510	5.400	0.110	5.662	5.498	0.164
5-CH	6.117	5.780	** 0.337 **	6.133	5.738	** 0.395 **
11-CH	7.253	6.881	** 0.337 **	7.280	6.955	** 0.325 **
9-CH	7.462	7.184	** 0.278 **	7.359	6.961	** 0.398 **
14-CH	7.243	7.012	** 0.231 **	7.384	7.132	** 0.252 **
12-CH	7.330	7.167	0.163	7.660	7.225	** 0.435 **

^1^ Δ*δ* = *δ***_1_** − *δ***_1_***_+_***_3_** or Δ*δ* = *δ***_2_** − *δ***_2_***_+_***_3_**; positive values mean a low-frequency shift.

**Table 6 ijms-24-17445-t006:** The chemical shifts, δ (ppm), of H6/H8 and H1′ proton signals of free nicked decamer duplex **3** and upon interaction with compounds **1** and **2** in D_2_O buffer (25 mM NaCl/25 mM K_3_PO_4_ + TSPA-*d*_4_), pH 6.0, temp = 10 °C; c_1_ = c_3_ = 0.6 mM. The chemical shift changes, Δδ ^1^ (ppm), of H6/H8 and H1′ proton signals of nicked decamer duplex **3** upon interaction with compounds **1** and **2**.

	δ Free 3	δ 1 and 3	Δδ ^1^	δ 2 and 3	Δδ ^1^	δ Free 3	δ 1 and 3	Δδ ^1^	δ 2 and 3	Δδ ^1^
	H6/H8 of 3					H1′ of 3				
G1	8.099	8.099	0.000	8.099	0.000	6.022	6.022	0.000	6.017	0.005
C2	7.421	7.422	−0.001	7.418	0.003	5.663	5.666	−0.003	5.663	0.000
G3	7.984	7.980	0.004	7.974	0.010	6.110	6.099	0.011	6.088	** 0.022 **
T4	7.311	7.306	0.005	7.298	0.013	6.095	6.090	0.005	6.082	0.013
T5	7.464	7.465	−0.001	7.458	0.006	6.215	6.151	** 0.064 **	6.184	** 0.031 **
nick										
G6	7.672	7.652	** 0.020 **	7.694	** −0.022 **	5.830	5.795	** 0.035 **	5.736	** 0.095 **
T7	7.592	7.562	** 0.030 **	7.545	** 0.047 **	6.093	6.0922	0.001	6.080	0.013
C8	7.494	7.491	0.003	7.486	0.008	5.766	5.763	0.003	5.756	0.010
G9	7.963	7.963	0.000	7.963	0.000	5.904	5.904	0.000	5.897	0.007
C10	7.603	7.603	0.000	7.603	0.000	6.238	6.238	0.000	6.238	0.000
G11	8.099	8.099	0.000	8.099	0.000	6.022	6.022	0.000	6.016	0.006
C12	7.396	7.393	0.003	7.394	0.002	5.633	5.633	0.000	5.633	0.000
G13	7.946	7.941	0.005	7.940	0.006	5.690	5.693	−0.003	5.690	0.000
A14	8.177	8.170	0.007	8.155	** 0.022 **	6.261	6.257	0.004	6.241	** 0.020 **
C15	7.177	7.188	−0.011	7.166	0.011	5.560	5.610	** −0.050 **	5.628	** −0.068 **
A16	8.053	8.073	** −0.020 **	8.081	** −0.028 **	5.826	5.807	** 0.019 **	5.779	** 0.047 **
A17	8.192	8.197	−0.005	8.189	0.003	6.156	6.150	0.006	6.140	0.016
C18	7.182	7.184	−0.002	7.182	0.000	5.604	5.602	0.002	5.602	0.002
G19	7.895	7.893	0.002	7.893	0.002	5.904	5.904	0.000	5.897	0.007
C20	7.575	7.575	0.000	7.575	0.000	6.226	6.226	0.000	6.226	0.000

^1^ Δ*δ* = *δ*_free **3**_ − *δ***_1_***_+_***_3_** or Δ*δ* = *δ*_free **3**_ − *δ***_2_***_+_***_3_**; positive values mean a low-frequency shift.

**Table 7 ijms-24-17445-t007:** Docking analysis with AutoDock-GPU.

**Ligand**	**Calculated Free Energy of Binding FE (kcal/mol) and Inhibition Constant Ki (nM)**							
	**X-ray Single Structure**				**1000 MD-Derived Structures**			
	**1K4T**		**1T8I**		**1K4T**		**1T8I**	
	**FE**	**Ki**	**FE**	**Ki**	**FE**	**Ki**	**FE**	**Ki**
**1** (5*R*,20*S*)	−10.97	9.16	−10.23	31.91	−10.77 ± 0.51	12.86	−11.22 ± 0.70	5.95
**2** (5*S*,20*S*)	−10.40	23.85	−10.24	31.25	−10.03 ± 0.46	44.71	−10.91 ± 0.74	10.15
TPT ^1^	−11.81	2.22	−11.47	3.89	−10.15 ± 0.49	36.25	−10.96 ± 0.79	9.22
CPT ^1^	−11.57	3.29	−11.20	6.19	−10.38 ± 0.47	24.48	−10.95 ± 0.77	9.38
SN38 ^1^	−11.26	5.57	−10.69	14.61	−10.36 ± 0.48	25.42	−11.04 ± 0.76	8.08

^1^ The results are obtained from [23].

**Table 8 ijms-24-17445-t008:** Docking analysis for receptor structures obtained in MD simulations of ternary complexes compounds **1** and **2**/DNA/Topo I. The most strongly bound complexes from a given family are marked in bold.

Ligand	Calculated Free Energy of Binding (kcal/mol), Inhibition Constant Ki (nM), and Cluster Size CS.											
	Family Structure 1			Family Structure 2			Family Structure 3			Family Structure 4		
	Energy	Ki	CS	Energy	Ki	CS	Energy	Ki	CS	Energy	Ki	CS
**1** (5*R*,20*S*)	−12.03 ± 0.58	1.52	968	−11.85 ± 0.50	2.07	685	−11.83 ± 0.41	2.13	455	**−12.48** **± 0.55**	**0.71**	**1012**
**2** (5*S*,20*S*)	−11.23 ± 0.38	5.92	537	−11.69 ± 0.96	2.69	1322	**−12.17** **± 1.05**	**1.20**	**1120**	−11.42 ± 0.61	4.29	754
TPT ^1^	**−10.27** **± 0.40**	**23.60**	**1956**	−10.49 ± 0.47	25.36	1196	−10.04 ± 0.01	44.12	11	−9.98 ± 0.03	48.44	93
CPT ^1^	**−10.33** **± 0.41**	**27.03**	**2844**	−10.06 ± 0.05	42.20	168	−9.96 ± 0.01	50.49	22	−10.01 ± 0.01	46.19	19
SN38 ^1^	**−10.38** **± 0.41**	**24.54**	**2407**	−10.40 ± 0.35	27.79	1219	−10.25 ± 0.10	30.80	203	−10.04 ± 0.02	44.04	36

^1^ The results are obtained from [23].

**Table 9 ijms-24-17445-t009:** MM-PBSA and MM-GBSA results of the calculations for the ligand/DNA/Topo I ternary complexes. The most strongly bound complexes from a given family are marked in bold.

**Ligand/Family Structure**	**Energy (kcal/mol)**	
	**PBSA**	**GBSA**
Compound **1**/1	−40.02 ± 3.89	**−52.32 ± 2.40**
Compound **1**/2	−40.06 ± 1.28	−45.35 ± 1.29
Compound **1**/3	−37.26 ± 1.56	−40.32 ± 1.99
Compound **1**/4	**−44.87 ± 1.01**	−49.92 ± 1.27
Compound **2**/1	−39.32 ± 1.22	**−51.34 ± 1.63**
Compound **2**/2	−35.81 ± 1.45	−43.52 ± 1.33
Compound **2**/3	−40.15 ± 1.60	−46.60 ± 1.51
Compound **2**/4	**−46.34 ± 0.89**	−48.69 ± 0.69
TPT/1 ^1^	−39.21 ± 1.62	−48.03 ± 0.58
CPT/1 ^1^	−38.69 ± 3.81	−48.89 ± 3.73

^1^ The results are obtained from [23].

## Data Availability

The authors confirm that the data supporting the findings of this study are available within the article and its Appendix A.

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
