# Peer review of "New 5-Substituted SN38 Derivatives: A Stability Study and Interaction with Model Nicked DNA by NMR and Molecular Modeling Methods"

_ijms, 2023, doi:10.3390/ijms242417445_

Round 1

Reviewer 1 Report

Comments and Suggestions for Authors

In the present manuscript, titled “New 5-substituted SN38 derivatives: A Stability Study and Interaction with Model nicked DNA by NMR and Molecular Modeling Methods” Elżbieta Bednarek and co-workers, investigated the two new derivatives of 5-substituted SN-38, labeled as 5(R)-(N-pyrrolidinyl)methyl-7-ethyl-10-hydroxycamptothecin and its diastereomer 5(S). Through a combination of NMR spectroscopy and molecular modeling, the research explores their chemical and configuration stability, as well as aggregation behavior at different concentrations.  NMR experiments focused on the interaction of each diastereomer with a nicked decamer duplex. The study revealed that the interaction occurred precisely at the DNA strand break site. Molecular modeling methods were employed to further understand the interaction mode and generate molecular models of the DNA/ligand complexes. Therefore, I would like to recommend the submitted article for publication after addressing the following comments and details:

1.     In the introduction, in the effort to elucidate the pharmacological role of the new diastereoisomeric derivatives in the inhibition process of Topo I, it would be beneficial to provide a structural overview of Topo I.

2.     For the carbon 5 isomers of compounds 1 and 2, on which in vitro biological assays were conducted using various cancer cells and normal cells, it is advisable to specify the types of cancer cells employed for these assays. Additionally, I recommend reporting the IC50 values obtained from these assays.

Comments on the Quality of English Language

Minor editing of English language required

Reviewer 2 Report

Comments and Suggestions for Authors

In my opinion, the manuscript entitled New 5-substituted SN38 derivatives: A Stability Study and 2 Interaction with Model nicked DNA by NMR and Molecular 3 Modeling Methods by Bednarek et al., endeavors to ascertain the capacity of the C5 substituent, containing the nucleophilic nitrogen, to exert an influence on the binding affinity within the molecular complex of DNA/inhibitor. The introduction is well done, according to the correct state of the art, the materials and methods needs minor revision and the results and discussion are well emphasized and compared with other results.

I have some comments and suggestions, as follows:

1.     At the introduction part, the aim of the present research is quite unclear. Please introduce a phrase in order to better highlight the aim of the present paper. It is recommended to include the phrase in the end of the introduction part. 

2.     When an abbreviation is used for the first time in the text it should be mentioned under brackets. For instance, ECD, IC50

3.     Line 124: Additionally, no additional signals appeared - please use instead of additional the word supplementary in order to not repeat the same word.

4.     Line 488 – the format of the writing is not in line with the whole manuscript. Please check it.

5.     Lines 423-428; please better detailed the HPLC method in order to be reproduce by scientific in the field. For instance, the mobile phase, the flow rate, the total analysis time. Mass spectrometric detection was used?

Thank you!

Comments on the Quality of English Language

Minor editing of English language required
